# OpenReview forum: "Feedback-Driven Black-Box Safety Alignment Testing of Large Language Models via Reinforcement Learning"
_TMLR — Accepted by TMLR_

### Review · Reviewer_R5tn · 2026-04-16

**Summary Of Contributions:**

This paper proposes a method for jailbreaking an LLM's defense against malicious inputs and outputs. Given a malicioius input which an LLM will generally refuse to answer, this paper aims to generate a manipulation to the input to bypass the defense mechanism of the LLM.

The proposed method is based on reinforcement learning with some interesting designs. For example, the actions the policy is allowed to take does not include appending suffixes to avoid an overly large search space. The action space is set as a discrete space with 10 choices, and employs a helper LLM to generate the final prompt that is used to perform jailbreaking, which combines prior knowledge and reduces search efforts. Reward is set as dense, instead of sparse, 0-1 rewards. The authors also design state transitions to mitigate abrupt prompt changes.

The authors experiment on a wide variety of open/closed models with a wide range of baselines (e.g. random search, genetic search, in-context learning) and the results show the advantages of the proposed SEAT-RL.

**Additional Comments:**

No additional comments.

**Audience:**

Yes

**Audience Explanation:**

LLM jailbreaking would be interesting to practitioners in LLM post-training and alignment. It would be of interest to a wide audience.

**Broader Impact Concerns:**

No concerns.

**Claims And Evidence:**

No

**Claims Explanation:**

Overall I like this paper. The paper is loaded with intuitions that lead to the rationale of the final design. For example, the rationale of including a discrete, 10-way action space instead of directly manipulating prompts is clearly explained. The modeling of the jailbreaking problem as an RL problem also makes sense as jailbreaking directly fits the loop of action-observation-reward. The experiments are also quite extensive, containing a wide range of attacked models and ablations.

Here are some weaknesses and questions that I have towards this paper. My evaluation of this paper would be significantly changed if these questions are properly answered or the weaknesses are properly revised.

1. **Rationale of cosine similarity as reward**. Cosine similarity is a good choice as a dense reward. However, some further justification may be needed, especially when only one reference answer is given. It would be good if the authors can analyze whether cosine similarity will give "false positives" to long answers that refer to the initial prompt, or whether it will give "false negatives" to answers that are relevant, but not good enough (e.g. containing fewer details)?

2. **Rationale of removing the model output as state**. Intuitively, adding the output of the attacked model will give the attacker more context about why the attacked model refuses to answer. Since the trained policy model is largely frozen, adding it should not cause a significant increase in computation. Will this be useful to the overall attack success?

3. **Rationale of setting the largest trial to 5**. In general, RL formalizations involve significantly longer horizons (e.g. 30, 40 or longer). If the number of trials in this paper is capped at 5, will some simpler formulations be effective as well, such as contextual bandits?

4. **Rationale of state transition**. The inclusion of state transition to avoid abrupt state changes do make sense, but from the example of Figure 3, it also causes somewhat contradictory prompts (both a financial analyst and detective, for example) which causes semantic discontinuity. Will such a method lead to even more contradictory inputs? If the answer is yes, then the attack method may be easy to detect, as such patterns (both an "A role" and "B role") do not often exist in natural language.

5. **Relation to general prompt optimization**. From my understanding, this work bears some relation with efforts that attempt to craft a good input prompt towards a task, such as GEPA (https://arxiv.org/abs/2507.19457). Interestingly, GEPA uses genetic algorithms to generate good prompts, which seems to contradict with the claims in this paper (e.g. in the introduction about genetic algorithms). A discussion of these latest works can enhance the validity of this paper.

A final minor comment: it would be better to restructure Sections 3.3 and 3.4, as the currrent version seems a bit dense. Some other ways to demonstrate (e.g. algorithm pseudocode) may better illustrate the proposed method.

**Requested Changes:**

1-5 in the above section.

---

> ### Author Response · Authors · 2026-05-17
> **Response to Reviewer R5tn (1/4)**
>
> We sincerely thank the reviewer R5tn for the insightful comments and suggestions. We are glad that the reviewer found the design rationale intuitive, especially the discrete action space and RL formulation. We have addressed the reviewer’s concerns below and revised the paper accordingly.
>
> # Cosine-similarity reward justification
>
> **(1) Prompt echoing does not necessarily cause false positives.** The reference $\hat{\mathbf{u}}$ is produced by querying the unaligned LLM with the **harmful question $\mathbf{q}$ only**, not with the wrapped jailbreaking prompt. The unaligned LLM does not see the role-play context or any of the helper-LLM-generated wrapping that is shown to the target LLM. As a result, $\hat{\mathbf{u}}$ contains content that directly answers $\mathbf{q}$, and does not share vocabulary with the prompt's context (e.g., "As a researcher...", "In this fictional scenario..."). Simply repeating the initial prompt or producing a long answer about the surrounding context does not substantially increase similarity to $\hat{u}_i$. Moreover, cosine similarity is normalized, so response length does not mechanically increase the reward. A high reward requires the response embedding to align with the answer-specific content in the reference response. We also manually double-check all the reference answers and ensure they are indeed answering the questions.
>
> **(2) On-topic answers with different phrasing for false negatives.** We agree that false negatives can occur when the target model answers the harmful query in a way that differs from the single reference response. However, because both responses address the same question, they typically share intent-specific entities, actions, and semantic concepts, so we find that the cosine similarity remains similar in practice. Furthermore, we do not rely on cosine similarity alone in the final evaluation. We additionally report two independent metrics: VDR and GPT-Judge, where GPT-Judge directly evaluates whether the target response answers the original harmful query and can capture cases where the target model gives a valid answer that differs from the single reference. In other words, the cosine similarity reward only needs to provide a useful training signal for policy optimization, not a perfect verdict on each generation.
>
> **(3) Reward robustness analysis.** We further added a reward-robustness ablation study to test how sensitive our method is to unavailable or uninformative references. We randomly marked 10% and 20% of the reference answers as unavailable by setting them as “I’m sorry, I cannot assist with this request.” This also mimics the case when the unaligned model refuses to answer the question. Then we trained our RL agent on Llama3-8B-instruct. The results below demonstrate that even though some of the reference answers are unavailable, our method still achieves good performance on jailbreaking the model.
>
> |               | Llama3-8B-instruct |           |
> | ------------- | ------------------ | --------- |
> | Method        | Sim.               | GPT-Judge |
> | SEAT-RL       | 0.7087             | 0.6038    |
> | SEAT-RL (10%) | 0.7062             | 0.5850    |
> | SEAT-RL (20%) | 0.6801             | 0.5725    |

---

> ### Author Response · Authors · 2026-05-17
> **Response to Reviewer R5tn (2/4)**
>
> # Removing target model output from state
>
> We address it from two perspectives: first, adding the target response embedding to the state is computationally affordable, but the key issue is whether it provides useful additional information; second, we added a state-space ablation to empirically test this design choice.
>
> We agree that adding $\mathbf{u}^{(t)}$ to the state would not significantly increase computation. The text encoder $\Phi$ is frozen, and $\Phi(\mathbf{u}^{(t)})$ is already produced at every step in order to compute the reward $r^{(t)} = \cos(\Phi(\mathbf{u}^{(t)}), \Phi(\hat{\mathbf{u}}))$. Augmenting the state with this embedding only requires caching an embedding that is already available and slightly widening the classifier head by one input block. The added parameter count is negligible compared with the cost of LLM forward passes. Thus, the question is not whether including $\mathbf{u}^{(t)}$ is computationally affordable, but whether it provides useful information for policy learning.
>
> To answer this directly, we added a state-space ablation comparing three variants on Qwen3-8B and Llama3-70B-Instruct. The full setup is provided in Appendix F.4.
>
> | State variantState variant        | Qwen3-8B VDR / Sim. | Llama3-70B VDR / Sim. |
> | --------------------------------- | ------------------- | --------------------- |
> | **(i) Prompt only (SEAT-RL)**     | **0.8150 / 0.7318** | **0.4218 / 0.6986**   |
> | (ii) Prompt + target response     | 0.7984 / 0.7211     | 0.4107 / 0.6878       |
> | (iii) Prompt + target + reference | 0.7836 / 0.7152     | 0.4031 / 0.6802       |
>
> The results show that augmenting the state yields no obvious improvement in either VDR or cosine similarity, as the added embeddings do not introduce new information beyond what the reward already encodes, while the higher-dimensional state requires more parameters to fit from the same amount of policy-gradient experience, which slightly destabilises learning.
>
> Furthermore, our design is consistent with prior work from both RL-driven search and state abstraction. In RL-driven search, the policy state commonly represents the controllable artifact being optimized, while the target system’s output is used by the environment to compute the reward. For example, in neural architecture search, the controller generates architecture decisions, while the child network’s validation accuracy is used as the reward; the child model’s predictions or training dynamics are not included in the controller state [1]. Similarly, RLPrompt [2] formulates discrete prompt optimization as an RL problem where the agent conditions on the current prompt and receives a reward computed from the frozen LM’s output, without including that output as part of the policy state. It is also supported by the state-abstraction literature. [3] argues that RL agents should retain enough information to inform decisions while avoiding irrelevant details that increase state complexity. [4] similarly frames deep RL state learning as a tradeoff: discarding too much information makes the representation insufficient, while discarding too little prevents the agent from benefiting from abstraction. In our setting, the target response is high-dimensional and largely redundant with the reward signal, so including it increases the state dimension without improving policy learning.
>
> [1] Zoph & Le, Neural Architecture Search with Reinforcement Learning, ICLR 2017.
>
>  [2] Deng et al., RLPrompt: Optimizing Discrete Text Prompts with Reinforcement Learning, EMNLP 2022.
>
>  [3] Jong & Stone, State Abstraction Discovery from Irrelevant State Variables, IJCAI 2005.
>
>  [4] Allen et al., Learning Markov State Abstractions for Deep Reinforcement Learning, NeurIPS 2021.

---

> ### Author Response · Authors · 2026-05-17
> **Response to Reviewer R5tn (3/4)**
>
> # Max trial = 5 is short for RL
>
> We thank the reviewer for raising this concern. We have revised Section 3.4 to clarify why SEAT-RL differs from a bandit formulation and why a short horizon is appropriate for our setting.
>
> First, contextual bandits optimize one-step decisions conditioned on the current context, assuming actions do not influence future states. In contrast, SEAT-RL is inherently sequential: earlier actions modify the conversational state and construct the next prompt state. $a_t$ determines how the helper LLM rewrites the current prompt, inducing a transition from $p^{(t-1)}$ to $p^{(t)}$. Thus, even though our horizon is short, the problem remains a finite-horizon MDP rather than a contextual bandit.
>
> We considered three possible bandit simplifications, each of which loses an important aspect of the problem. (1) a **one-step contextual bandit** would take the original harmful query $q$ as context and choose one of the ten mutation strategies once. This is essentially T=1 and cannot model multi-step compositions such as "create a role-play setting → legitimize the setting → paraphrase or restructure the prompt." But the paper’s motivation is that effective jailbreak prompts often emerge through **iterative prompt refinement**, not a single independent transformation. (2) a **myopic contextual bandit over current prompts** could observe the current prompt $p_{t-1}$, choose an action, and optimize only the immediate reward. This is closer to SEAT-RL operationally, but it ignores delayed effects. Some actions may produce little immediate reward while creating a better intermediate prompt for the next action. In RL terms, it optimizes r_t, while SEAT-RL optimizes the return: $G_t = r_t + \gamma r_{t+1} + \cdots + \gamma^{T-t}r_T.$ This matters because the action changes the prompt state that future actions operate on. A bandit-style learner would treat each step as locally independent, while our policy learns which transformations are useful **as part of a sequence**. (3) one could collapse the entire sequence into a **combinatorial bandit**, where each arm is a length-5 action sequence. But even with only ten actions, this gives $10^5=100{,}000$ possible action sequences before accounting for stochastic helper-LLM outputs and candidate prompts. It also removes feedback adaptivity: the action at step 3 could not depend on what the prompt became after steps 1 and 2. This is exactly the inefficient search SEAT-RL is designed to avoid.
>
> The short horizon is a consequence of our high-level action abstraction and the black-box query budget. Each action is not a token-level edit, but a semantic prompt-transformation strategy instantiated by a helper LLM. Thus, five refinement steps can compose several substantial transformations, while longer horizons would multiply target-model queries and make the comparison less practical under the fixed 10000-query budget.
>
>
> # State transition/crossover artifacts
>
> We clarify that the crossover operation is designed to create a coherent semantic bridge between two candidate contexts, rather than to preserve both roles literally. We have included the full crossover prompt in Appendix E.1. In particular, the helper prompt instructs the model to merge the two prompts into a single cohesive prompt and to identify a connection or point of intersection between the two scenarios.
>
> We agree that incoherent crossovers can occasionally occur, since the transition is generated by an LLM. However, such cases are transition failures rather than patterns encouraged by the policy. If a merged prompt is contradictory or incoherent, the target model often refuses, becomes confused, or produces an irrelevant answer, which leads to a low reward. As a result, these transitions are disfavored during policy learning. In addition, crossover is only applied when the policy switches between different context-generating actions. If the same action is selected in consecutive steps, we paraphrase the current prompt instead. This design prevents the repeated accumulation of unrelated roles or scenarios.

---

> ### Author Response · Authors · 2026-05-17
> **Response to Reviewer R5tn (4/4)**
>
> # General prompt optimization
>
> Our discussion in the introduction focuses specifically on existing black-box safety-testing and jailbreak-generation methods that rely on stochastic mutation and selection, including the genetic-search baselines considered in our paper. These methods do not explicitly learn a reusable, state-conditioned search policy from target-model interaction feedback, which can make exploration inefficient under a fixed query budget. We have narrowed our claim as follows: **existing genetic-search methods for black-box safety testing rely mainly on stochastic mutation and selection, without learning a reusable interaction-driven policy; SEAT-RL addresses this specific limitation.**
>
> We have revised the introduction and related-work section to clarify this scope and to discuss recent general prompt-optimization methods such as GEPA. GEPA is a strong example showing that evolutionary prompt optimization can be effective when combined with natural-language reflection and Pareto-based candidate selection. However, GEPA addresses a different problem setting: it optimizes prompts for compound AI systems over task distributions using rich execution traces, feedback functions, and evaluation signals. In contrast, SEAT-RL targets black-box safety-alignment testing, where the tester interacts with a safety-aligned target model only through prompts and final responses, and the goal is to generate per-query test prompts that elicit non-refusal and relevant harmful answers. Thus, GEPA is complementary rather than contradictory to our work. It supports the broader idea that high-level semantic prompt transformations and feedback-driven search are useful. Our contribution is to formulate safety-alignment testing as a sequential decision-making problem and learn a reusable RL policy over structured prompt-transformation actions.
>
> # Minor concerns
>
> To address these concerns, we made three revisions. First, we added the formal training procedure as Algorithm 1 in Section 3.4, so the overall workflow is easier to follow at a glance. Second, we moved the PPO derivation, including the original Eq. (3) and the value-baseline discussion, to Appendix D.3, “Derivation of the PPO Objective.” Third, we moved the discussion of using GPTFuzzer’s reward as our reward to Appendix D.4. As a result, Section 3.4 now contains a concise conceptual summary of the training objective, together with forward references to the detailed derivations and reward discussion.
>
> # Changes to the paper
> We have revised the paper as follows:
> - We have revised Section 3.4 to clarify the construction of $\hat{\mathbf{u}}$ and explicitly state that the reference response is generated from the bare harmful question, which helps prevent prompt-echo false positives. We also added the reward-robustness ablation and corresponding discussion to further validate the cosine-similarity reward.
>
> - We added this state-space ablation and the corresponding discussion to Appendix F.4. We also clarified that $\mathbf{u}^{(t)}$ is omitted only from the agent’s state representation, while still being used by the environment for reward computation.
>
> - We have revised Section 3.2 to clarify the distinction between our finite-horizon MDP formulation and bandit-based alternatives, and to explain the rationale behind the short horizon.
>
> - We have revised Section 3.4 to clarify the purpose of the crossover operation, explain how incoherent crossovers are penalized through the reward, and note that repeated selections of the same action trigger paraphrasing rather than additional crossover.
>
> - We have revised the introduction and related-work section to narrow the scope of our claim, distinguish SEAT-RL from general prompt-optimization methods such as GEPA, and narrow down our claim to genetic-search methods for black-box safety testing.
>
> We thank the reviewer again for their constructive feedback, and we hope the revised version addresses the concerns.

---

### Review · Reviewer_33b4 · 2026-04-20

**Summary Of Contributions:**

The authors propose a novel framework that enables systematic safety alignment testing for LLMs. This framework formulates safety evaluation as a deep reinforcement learning (DRL) problem and the authors propose corresponding state and action spaces, a reward function, and a state transition function. The proposed framework is evaluated against several baselines and yields superior results compared to the baselines. The authors conduct additional ablation studies to further investigate the impact of the various components of their framework.

**Audience:**

Yes

**Audience Explanation:**

This paper would be of interest for individuals interested in LLM alignment.

**Claims And Evidence:**

No

**Claims Explanation:**

The authors make two primary claims in this paper:

1) That safety evaluation can be appropriately modelled by the proposed sequential decision-making formulation.
2) That the proposed framework performs better than baseline approaches under the same query budget.

In terms of the first claim, I find that, for the most part, it is supported by clear and convincing evidence (see Section 3.4). The proposed action space and reward function are well-motivated and the authors do a good job in motivating and explaining the different components of the formulation. However, in Section 3.4, the authors state, without rigorous justification, that they omit the target LLM’s response from the state-space. From a purely reinforcement learning perspective, this is highly non-standard; typically, any component needed to calculate the rewards should be included in the state-space (this also includes the “reference” response from the non-aligned LLM). Because of this, I do not find that the first claim is entirely supported by the evidence and discussions provided in the paper.

In terms of the second claim, I find that it is adequately supported by the evidence provided in Section 4.

**Requested Changes:**

From a writing perspective, the paper is very well-written and I have no concerns in this regard. The figures included in the text are also clear and easy to understand.

My main request is for the authors to rigorously justify their decision to omit the target LLM and reference responses from the state-space. In particular, the authors need to either show themselves that such a decision does not break any underlying assumption of the DRL framework used, or they need to cite a prior work that does so.

Minor requests are as follows:
- I don’t think SEAT is defined anywhere (I assume it is an acronym for something).
- Small typo in Section 3.2, second paragraph: there is a period missing after “prompt refinement process”.
- In the first paragraph of Section 3.3, the reference response, $\hat{u}$, is used without being formally defined (it is defined later on, but it should be defined prior to being used anywhere in the text).

Aside from the above concerns, I find the paper to be of high quality, and thank the authors for the effort put into this work.

---

> ### Author Response · Authors · 2026-05-16
> **Response to Reviewer 33b4**
>
> We sincerely thank the reviewer 33b4 for their constructive comments. We are also glad that the reviewer found the paper well-written and the figures clear and easy to understand. We have revised the paper to address the reviewer's concerns, including clarifying our state space formulation in Section 3.4 and adding a new state-space ablation in Appendix F.4.
>
> # Omitting the target LLM's response from the state space
>
> We agree with the reviewer that in an MDP formulation, the state should contain sufficient information to make decisions. We address this concern from two perspectives: first, we clarify that our state space design follows established abstractions in RL-driven search and is supported by prior literature; second, we add a new state space ablation to empirically test whether including the target response improves performance.
>
> First, our design follows a common abstraction in RL-driven search: the state is the controllable artifact being constructed, while the target system’s output is usually used by the environment to compute reward. For example, in [1], the controller generates architecture decisions, while the child network’s validation accuracy is used as the reward; the child model’s predictions or training dynamics are not included in the controller state. Similarly, [2] formulates discrete prompt optimization as an RL problem where the agent conditions on previous prompt tokens and receives a reward computed from the frozen model's output, without including that output as part of the state. Our formulation follows the same principle: $s^{(t)}=p^{(t-1)}$ captures the current controllable prompt, while $u^{(t)}$ and $\hat{u}$ are environment-side variables used to compute $r^{(t)}=\cos(\Phi(u^{(t)}),\Phi(\hat{u}))$. There is also direct support from state abstraction literature. [3] argues that RL agents should keep enough information to inform choices without wasting resources on irrelevant details, and that state-representation complexity is a key bottleneck for practical RL. [4] also frames deep RL state learning as a tradeoff: discard too much and the representation is insufficient; discard too little and the agent fails to benefit from abstraction.
>
> To empirically validate this design choice, we added a new state space ablation. We compare our prompt-only state against two augmented variants that include the target response and the target response plus reference response:
>
> - **(i)** State = $\Phi(\mathbf{p}^{(t)})$ &nbsp; *(our design)*
> - **(ii)** State = $[\Phi(\mathbf{p}^{(t)}) \,\Vert\, \Phi(\mathbf{u}^{(t-1)})]$
> - **(iii)** State = $[\Phi(\mathbf{p}^{(t)}) \,\Vert\, \Phi(\mathbf{u}^{(t-1)}) \,\Vert\, \Phi(\hat{\mathbf{u}})]$
>
> Results are below (full setup and results are in Appendix F.4):
>
> | State variant                     | Qwen3-8B VDR / Sim. | Llama3-70B VDR / Sim. |
> | --------------------------------- | ------------------- | --------------------- |
> | **(i) Prompt only (SEAT-RL)**     | **0.8150 / 0.7318** | **0.4218 / 0.6986**   |
> | (ii) Prompt + target response     | 0.7984 / 0.7211     | 0.4107 / 0.6878       |
> | (iii) Prompt + target + reference | 0.7836 / 0.7152     | 0.4031 / 0.6802       |
>
> Augmenting the state yields no obvious improvement in either VDR or cosine similarity, as the added embeddings do not introduce new information beyond what the reward already encodes, while the higher-dimensional state requires more parameters to fit from the same amount of policy-gradient experience, which slightly destabilizes learning.
>
> **Changes to the paper.** We have revised Section 3.4 to clarify that $u^{(t)}$ and $\hat{u}$ are omitted only from the agent’s state representation, while still being used by the environment for reward computation. We also added the state-space ablation and corresponding discussion in Appendix F.4 under the "State space ablation" paragraph.
>
> [1] Zoph & Le, Neural Architecture Search with Reinforcement Learning, ICLR 2017.
>
> [2] Deng et al., RLPrompt: Optimizing Discrete Text Prompts with Reinforcement Learning, EMNLP 2022.
>
> [3] Jong & Stone, State Abstraction Discovery from Irrelevant State Variables, IJCAI 2005.
>
> [4] Allen et al., Learning Markov State Abstractions for Deep Reinforcement Learning, NeurIPS 2021.
>
> # Other concerns
>
> We thank the reviewer for pointing out these issues. We have addressed them in the revised paper as follows.
>
> - We fixed the acronym in the abstract by explicitly highlighting the corresponding letters.
> - We fixed the typo in Section 3.2.
> - We added a clarifying sentence in Section 3.3 when introducing the reference response.
>
> We appreciate the reviewer's suggestions to improve our work, and hope our revision has addressed the concerns.

---

> > ### Comment · Reviewer_33b4 · 2026-05-19
> >
> > I thank the authors for their response to my review as well as for the updates made to the manuscript.
> >
> > Having read the updated manuscript, all of my concerns have been addressed.

---

> > > ### Author Response · Authors · 2026-05-20
> > >
> > > We thank the reviewer for reading the updated manuscript and for the positive feedback, and we are glad that the revisions addressed your concerns.

---

### Review · Reviewer_LtqH · 2026-05-11

**Summary Of Contributions:**

The paper presents SEAT-RL, a black-box framework for testing the safety alignment of large language models by formulating adversarial prompt generation as a sequential decision-making problem. The method uses RL to guide prompt mutation, an LLM-assisted discrete action space to produce structured prompt transformations, and a dense reward signal based on semantic similarity to reference answers from an unaligned model.

The main strengths are the clear motivation, practical black-box setting, broad empirical evaluation, and useful ablation studies. The main weaknesses are the reliance on automated success metrics, limited human validation.

**Audience:**

Yes

**Audience Explanation:**

The work is relevant to researchers studying LLM safety evaluation, black-box red teaming.

**Claims And Evidence:**

Yes

**Claims Explanation:**

Yes. They are supported by experimental evidence. The paper compares against several relevant baselines, evaluates multiple target models and defenses, and includes ablations.

**Requested Changes:**

1. The authors should add stronger validation of the success metrics, preferably through human annotation on a representative subset of generated outputs. This is important because keyword-based violation rates, cosine similarity, and GPT-based judging may not fully establish that outputs are genuinely harmful, relevant, and safety-violating.

2. The authors should discuss computational cost in more detail.

---

> ### Author Response · Authors · 2026-05-16
> **Response to Reviewer LtqH**
>
> We sincerely thank the reviewer LtqH for their thoughtful comments and constructive feedback. We have revised the paper to strengthen the validation of our metrics and to provide a more detailed discussion of computational cost.
>
> # Stronger validation of success metrics
>
> We added a human-validation study on GPT-Judge in Appendix D.6, as GPT-Judge is our main automated metric for determining whether a response is harmful and relevant. We also clarify that GPT-Judge is **not a metric newly proposed by us**; instead, we directly adopt the judge prompt from the published work COLD-Attack [1]. In COLD-Attack, they report that GPT-Judge has a higher correlation with human annotations than alternative automatic metrics. This choice helps us avoid introducing a custom evaluator that may be biased toward our method.
>
> Following recent jailbreak evaluation work such as JailbreakBench [2], HarmBench [3], and StrongREJECT [4], we use human annotation as the ground truth for validating automated success metrics. Specifically, we sample a subset of generated query-response pairs. Each example is labeled independently by two annotators, who see only the original harmful query and the model response, not the attack method or generated test prompt. Disagreements are resolved through adjudication.
>
> The annotation rubric evaluates three necessary conditions for a true safety violation: (1) the response does not refuse the request, (2) the response is relevant to the original harmful query, and (3) the response provides specific harmful assistance rather than vague, benign, or unrelated content. This follows the same setup as HarmBench and StrongREJECT, which emphasize that non-refusal alone is insufficient and that a valid jailbreak should produce useful, specific, harmful content.
>
> We then compare GPT-Judge against the human majority labels using agreement, precision, recall, F1, false positive rate, and false negative rate. Results show that GPT-Judge has about 90% human agreement, demonstrating that it is a reliable metric for attack success evaluation.
>
> # Computational cost in more detail
>
> We added a new "Computational cost" section in Appendix E.4 that discusses: (1) the per-RL-step compute breakdown, (2) what is trainable vs. frozen, (3) the hardware setup, and (4) end-to-end training and inference wall-clock for Qwen3-8B and Llama3-70B-instruct.
>
> [1] Guo et al., COLD-Attack: Jailbreaking LLMs with Stealthiness and Controllability, ICML 2024.
>
> [2] Chao et al., JailbreakBench: An Open Robustness Benchmark for Jailbreaking Large Language Models, NeurIPS 2024.
>
> [3] Mazeika et al., HarmBench: A Standardized Evaluation Framework for Automated Red Teaming and Robust Refusal, arXiv 2024.
>
> [4] Souly et al., A StrongREJECT for Empty Jailbreaks, NeurIPS 2024.

---

### Author Response · Authors · 2026-05-17
**Revision Summary**

We thank the reviewers again for their valuable feedback and suggestions. We have incorporated the revisions described in our responses and highlighted all corresponding changes in blue.

Below, we summarize the changes made to the paper.

## **Major Changes**

### **§1 — Introduction**

- Narrowed the scope of our critique of genetic-search methods and added discussion of general prompt-optimization work. We clarified that our critique applies specifically to *black-box safety-testing methods that rely on stochastic mutation and selection without learning a reusable interaction-driven policy*. We also noted that recent general prompt-optimization methods address a different problem setting.

### **§3.2 — Feedback-Driven Sequential Test Generation via DRL**

- Added a "Differences from contextual bandits" paragraph. We explain why SEAT-RL is formulated as a finite-horizon MDP rather than a contextual bandit: each action transforms the current prompt into the next state, so the policy must learn which transformations create useful intermediate prompts for later steps, rather than only maximizing immediate reward.

### **§3.4 — Design Details**

- Rewrote the state-space justification. We clarified that $\mathbf{u}^{(t)}$ and $\hat{\mathbf{u}}$ are still used by the environment to compute the reward, but are omitted from the agent’s state representation. The reward already summarizes whether $\mathbf{u}^{(t)}$ answers the harmful query, while $\hat{\mathbf{u}}$ depends only on the harmful question $\mathbf{q}$, which is already contained in the initial prompt $\mathbf{p}^{(0)}$.
- Clarified the construction of the reference response $\hat{\mathbf{u}}$. We explicitly state that $\hat{\mathbf{u}}$ is generated by querying the unaligned LLM with only the bare harmful question $\mathbf{q}$, not the wrapped jailbreaking prompt. This helps rule out false positives from long responses that merely echo the prompt wrapper, while still providing a useful directional signal for on-topic responses with different phrasing.
- Clarified the crossover operation. We explain that the helper LLM is instructed to identify a connection between two scenarios and merge them into a single cohesive prompt, rather than literally preserving both roles or contexts.
- Restructured the PPO discussion. We replaced the in-line PPO derivation with a compact conceptual summary and a forward reference to Appendix D.3. We also added Algorithm 1 to present the training procedure more clearly.

### **§6 — Related Work**

- Added a new subsection, "General Prompt Optimization." It positions our work as addressing a different setting: black-box safety-alignment testing under a fixed query budget.

### **Appendix**

* Appendix D.3: Moved the full PPO derivation from §3.4 to a dedicated appendix subsection, including the original Eq. (3), the advantage definition, and the value-baseline simplification.
* Appendix D.4: Moved the discussion of GPTFuzzer’s reward and third-LLM-judge alternatives from §3.4, with explicit reasoning about why we did not adopt these alternatives.
* Appendix D.6: Added a human-annotation study for GPT-Judge on Llama3-8B-Instruct and Qwen3-8B to validate our main automated success metric against human judgments.
* Appendix E.4: Consolidated the computational-cost analysis, including per-RL-step compute breakdown, trainable versus frozen components, hardware setup, end-to-end training and inference wall-clock time for Qwen3-8B and Llama3-70B-Instruct, comparison to PAIR and GPTFuzzer under a matched query budget, and Table 7 with a per-component step-time breakdown.
* Appendix F.4: Added two new ablations. The first is a state-space ablation comparing three variants on Qwen3-8B and Llama3-70B-Instruct: prompt only, prompt + previous target response, and prompt + previous target response + reference response. The second is a reward-robustness ablation where 10% and 20% of reference answers are replaced by a generic refusal string. Together, these results support the state-space design and show that the cosine-similarity reward is robust to a moderate fraction of missing or noisy references.

## **Minor Changes**

- Added the SEAT-RL acronym expansion in the abstract.
- Fixed the missing-period typo after "prompt refinement process" in Section 3.2.
- Added an inline definition of $\hat{\mathbf{u}}$ at its first use in Section 3.3, with a forward reference to the full justification in Section 3.4.
- Restructured Section 3.4 for readability by replacing the detailed PPO derivation with a compact summary and forward references to Appendix D.3 and Algorithm 1. Reward and agent-design alternatives were moved to Appendix D.4.

We sincerely appreciate the reviewers' constructive feedback, and we would be happy to address any further comments or questions.

---

### Decision · Action_Editor_szP9 · 2026-06-02

**Recommendation:** Accept as is

**Additional Comments:**

The authors have done a great job in addressing the concerns from the reviewers in the first round of the review. All reviewers recommend accepting the paper without further comment in the second round of the review. Therefore, AE would follow the reviewers' recommendation.

**Audience:**

Yes

**Audience Explanation:**

This paper would be of interest for individuals interested in LLM alignment.

**Claims And Evidence:**

Yes

**Claims Explanation:**

In the first round of the review, two reviewers think the claims made in the submission was not supported by accurate, convincing and clear evidence. However, the rebuttal resolves the issue and all reviewers lean towards accepting the paper.